# An Alternative Cell Therapy for Cancers: Induced Pluripotent Stem Cell (iPSC)-Derived Natural Killer Cells

**DOI:** 10.3390/biomedicines9101323

**Published:** 2021-09-26

**Authors:** Li-Jie Hsu, Chao-Lin Liu, Ming-Ling Kuo, Chia-Ning Shen, Chia-Rui Shen

**Affiliations:** 1Department of Medical Biotechnology and Laboratory Science, College of Medicine, Chang Gung University, Taoyuan 333, Taiwan; D0815001@cgu.edu.tw; 2PhD Program in Biotechnology Industry, College of Medicine, Chang Gung University, Taoyuan 333, Taiwan; 3Department of Chemical Engineering, Ming Chi University of Technology, New Taipei 243, Taiwan; clliu@mail.mcut.edu.tw; 4Biochemical Technology R&D Center, Ming Chi University of Technology, New Taipei 243, Taiwan; 5Department of Microbiology and Immunology, College of Medicine, Chang Gung University, Taoyuan 333, Taiwan; mingling@mail.cgu.edu.tw; 6Center of Molecular and Clinical Immunology, Chang Gung University, Taoyuan 333, Taiwan; 7Division of Allergy, Asthma, and Rheumatology, Department of Pediatrics, Lin-Kou Chang Gung Memorial Hospital, Taoyuan 333, Taiwan; 8Department of Pediatrics, New Taipei Municipal TuCheng Hospital, New Taipei 236, Taiwan; 9Genomics Research Center, Academia Sinica, Taipei 115, Taiwan; cnshen@gate.sinica.edu.tw; 10Department of Ophthalmology, Lin-Kou Chang Gung Memorial Hospital, Taoyuan 333, Taiwan

**Keywords:** induced pluripotent stem cells (iPSCs), natural killer cells (NK cells), cell therapy

## Abstract

Cell therapy is usually defined as the treatment or prevention of human disease by supplementation with cells that have been selected, manipulated, and pharmacologically treated or altered outside the body (ex vivo). Induced pluripotent stem cells (iPSCs), with their unique characteristics of indefinite expansion in cultures and genetic modifications, represent an ideal cell source for differentiation into specialized cell types. Cell therapy has recently become one of the most promising therapeutic approaches for cancers, and different immune cell types are selected as therapeutic platforms. Natural killer (NK) cells are shown to be effective tumor cell killers and do not cause graft-vs-host disease (GVHD), making them excellent candidates for, and facilitating the development of, “off-the-shelf” cell therapies. In this review, we summarize the progress in the past decade in the advent of iPSC technology and review recent developments in gene-modified iPSC-NK cells as readily available “off-the-shelf” cellular therapies.

## 1. Introduction

Cell therapy (also called cellular therapy or cytotherapy) is defined as therapy in which cellular material is injected into a patient. It is a technology that relies on replacing diseased or dysfunctional cells with healthy, functioning ones. This idea was initialized in 1931 when Paul Niehans (1882–1971) attempted to cure a patient by injecting material from calf embryos or harvesting cells from young animals or the fetus to treat severely ill patients [1]. Therefore, he is regarded as the inventor of cell therapy even though his claims have never been validated. In fact, this major breakthrough in stem cell research is considered the discovery of hematopoietic stem cells (HSCs), which are broadly applied for treating hematological cancers and various disorders of the blood and immune system in the clinic [2]. Before 2006, people believed that the cells were divided from the “organ-specific stem cells” of the tissue in which they resided, and that the cell types were also very limited. In addition, these cells may have irreversibly lost the capacity to generate other cell types. In 2006, Shinya Yamanaka and Kazutoshi Takahashi introduced induced pluripotent stem cells (iPSCs) [3], which can stably proliferate and serve as an unlimited source of cells to circumvent the original restriction. In the classification of stem cells, iPSCs are descendants of totipotent cells [4] and can be sufficiently expanded for transplantation use and disease treatment.

Currently, cell therapy can be divided into two categories (Figure 1): (1) stem cell therapy, including HSCs, mesenchymal stem cells (MSCs), iPSCs, adult stem cells and, most controversially, embryonic stem cells; and (2) immune cell therapy, via cell-mediated immunity, by transplanting macrophages [5], T cells [6], dendritic cells (DCs) [7], or NK cells [8] into patients to fight cancer cells [9]. iPSC technology has evolved rapidly and offers new perspectives on the production of immunotherapeutic cellular products. The generation of safe master iPSC lines bearing genetic modifications that confer the desired characteristics of the final product can facilitate the development of “off-the-shelf” cellular therapeutics for more patients and types of malignancy (breast cancer, neuroblastoma, epithelial tumors, melanoma) [10].

Immunotherapy has become a cornerstone in cell therapy and is an innovative approach for the treatment of cancer. The first FDA-approved gene-edited T-cell products (chimeric antigen receptor-modified T, CAR-T) for lymphoma and leukemia came out on the market in 2017 [11]. Recently, CAR-T cell therapy has been successful and become a clinical hotspot in tumor immunotherapy. However, a key challenge for the wider implementation of cell therapy concerns the laborious procedures of identifying HLA-matched healthy related or unrelated donors and harvesting their cells for engineering and infusion into one patient. Additionally, this application is limited by inherent risks such as graft-versus-host disease (GvHD), cytokine release syndrome (CRS), and immune effector cell–associated neurotoxicity syndrome (ICANS) [12]. The process is very lengthy and cumbersome [13,14]. Thus, the first clinical trial by Ruggeri et al., who proposed the potent antitumor efficacy of allogeneic NK cells, was performed in the context of hematopoietic stem cell transplantation (HSCT) [15], and the results revealed the potential realization of off-the-shelf products, making CAR NK cell therapies universal products, which might have a better safety profile than CAR-T cell therapy.

At present, it is known that NK cells not only detect and identify malignant cancer cells but also induce cancer cell death and even help trigger a broader adaptive immune response to fully engage and fight tumor cells. The safety of NK cell-based therapies is demonstrated in both autologous and allogeneic haploidentical settings [16,17,18,19]. Clinical studies show that NK cells are cytotoxic against a wide range of solid cancer tumor cells in vitro. The antitumor activities of adoptively transferred NK cells in vivo have also been demonstrated in preclinical xenograft mouse models of ovarian cancer, glioblastoma, and metastatic colorectal cancer [20]. NK cell-based immunotherapy has emerged as a promising therapeutic approach for hematological malignancies and solid tumors. Currently, NK cells can be derived from autologous or allogeneic sources, such as peripheral blood (PB), and can also be differentiated from induced iPSCs and HSCs [21]. Similar to T cells, NK cells can be engineered to better recognize a specific tumor. However, they have some advantages: they can detect a greater number of chemical signals from tumors than T cells; they are less prone to attack healthy tissues than T cells are. Thus, NK cell therapy could complement, and in some scenarios substitute for, T cell-based adoptive therapies to maximize antitumor effects and reduce treatment toxicity [22].

## 2. Induced Pluripotent Stem Cells (iPSCs)

In 2006, Shinya Yamanaka and Kazutoshi Takahashi successfully developed mouse iPSCs by using a retrovirus to deliver “Yamanaka factors” into somatic cells (mouse fibroblasts) [3]. The term “Yamanaka factors” means the combination of four reprogramming transcription factors—Oct3/4, Sox2, c-Myc, and Klf4 [23]—with the capacity to indefinitely propagate in vitro and the ability to differentiate into all somatic cell types upon receiving environmental cues. One year later, Shinya Yamanaka successfully generated iPSCs from human fibroblasts [24]. However, since c-Myc and Klf4 are oncogenes, they increase the risk of chromosomal instability and tumorigenesis. Thereafter, numerous studies have focused on identifying different reprogramming factors [3,24,25]. For example, Nanog and LIN28 can replace Klf4 and c-Myc, respectively [26]. Additionally, estrogen-related receptor beta (ESRRβ) can replace Klf4 [27]. Currently, iPSCs can be routinely generated from a variety of easily obtainable sources, such as skin and PB [24,28], and employ a combination of different reprogramming factors [24,29,30,31,32]. Indeed, a variety of studies have demonstrated the combination of different reprogramming factors utilized in a variety of cell types for the generation of iPSCs [3,24,27,28,29,30,31,32,33,34,35,36,37,38,39].

Moreover, scientists are investigating the mechanisms of those transcription factors involved in the generation of iPSCs [27,31,33,40]. Currently, it is well known that Oct4, Sox2, and Nanog, when bound together, activate the promoters of both genes (Sox2 and Oct4) and subsequently enhance the stability of pluripotency gene expression [32,41]. In fact, Sox2 and Oct4 have attracted attention since the discovery that these genes play critical roles during embryogenesis [42]. Sox2, which is a high-mobility group DNA-binding domain transcription factor, is essential for early embryogenesis in mice [43]. The increased expression (~2-fold) of Sox2 in embryonic stem cells (ESCs) induces ESC differentiation into cells that express markers of ectoderm and mesoderm, but not endoderm [44]. Oct4 is highly expressed in pluripotent cells and becomes silenced upon differentiation [45], such as by Oct4-deficient embryos, which fail to form an inner cell mass [46]. However, if one of these transcription factors is utilized, no function is detected until they are grouped together in complexes composed of a wide array of other proteins [42].

### 2.1. Strategies for Generating iPSCs

Although reprogramming is inefficient and tedious (the reported range is 0.00002~1% in different laboratories), technological advances have led to the tremendous development of nonintegrated viruses [47,48,49,50,51,52,53,54] and nonviral methods [55,56,57,58]. The nonintegrated methods include episomal DNA [47], adenovirus [48], Sendai virus [49], piggyBac (PB) transposons [50], small circles [51], recombinant proteins [52], synthetically modified mRNA [53] and microRNAs [54]. Among them, free DNA, synthetic mRNA, and Sendai viruses are commonly used to derive unintegrated iPSCs due to their relative simplicity, high efficiency, and elimination of insertional mutagenesis and transgene reactivation [58]. Nonviral methods, including plasmid transfection [55], minicircle vectors [56], transposon vectors [57], and liposomal magnetofection (LMF) [56] are also relatively safe due to the absence of, or minimal, integration possibilities, but are limited by their low efficiency and slow kinetics. Therefore, these techniques are improved to overcome various bottlenecks for their efficiency [59]. In summary, they are all able to undergo essentially unlimited expansion in vitro without losing pluripotency.

It appears that reprogramming towards iPSCs also benefits the introduction of genes into iPSCs and enables the correction of disease-causing gene mutations in patient-derived iPSCs or the delivery of specific mutations into non-disease-affected wild-type iPSCs. This process helps to create iPSC-based disease cell models, which could be beneficial for new drug screenings [60]. iPSC-based drug screening platforms are especially helpful for complicated or unintelligible diseases such as Parkinson’s disease [61,62], Alzheimer’s disease [63,64] and spinal cord injuries [65,66]. Figure 2 summarizes the generation of iPSCs and applications. 

Gene editing technology is a next-generation stem cell therapy. Nuclease-based gene editing systems, including zinc-finger nucleases (ZFNs), transcription activator-like effector nucleases (TALENs) and the CRISPR-Cas9 system, are among the most commonly used [67]. CRISPR–Cas9 technology in particular has attracted much attention and gained wide usage in the gene editing of human ESCs [68,69] and iPSCs [70,71], owing to its simplicity of design and ease of use. It is defined by a guide RNA that binds to the Cas nuclease and facilitates the design of new targeting constructs [72]. Unlike CRISPR–Cas9, ZFN and TALEN systems require the dimerization of the attached Fok1 endonuclease to induce their targeted double strand breaks (DSBs) [73]. However, the above technologies encounter several challenges. The major challenge is the “off-target effect”, which may induce unintended responses, leading to the risk of disaster damage. The societal challenge is how to mitigate the sense of uncertainty and fear of catastrophic misuse [74]. The first clinical trials (NCT03655678 and CT03745287) involving ex vivo “CRISPR–Cas9 genome editing in HSCs” for the treatment of hemoglobinopathies were initiated in 2018 [75,76]. Since off-target effects remain a challenge for the clinical translation of CRISPR, it is important to develop ultrasensitive methods to identify them. “Prime editing” based on RNA uses a modified Cas9 coupled to an engineered reverse transcriptase to make precise modifications in genomic sequences without creating DSBs [77,78,79]. The precision and versatility of prime editing may allow it to rival the CRISPR system as the preferred genomic editing tool [67]. 

### 2.2. Clinical Application of iPSC-Derived Products

Cell therapy appears to be the best or most common alternative choice of available treatments for some diseases. However, immune rejection is the major challenge. Due to the autologous cell source, this is a privilege of iPSC-derived products in regenerative medicine. Autologous iPSCs are immunologically identical to the host and can be established from host somatic cells and differentiated into a variety of cell types for cell therapy, including immune cells such as natural killer (NK) cells and macrophages. The bank for storing iPSC cell lines was established according to donors with identified human leukocyte antigen (HLA) and has become an effective way to provide recipients with the maximum number of HLA-matched cell products derived from iPSC lines [80].

In 2014, the world’s first clinical trial in Japan was performed based on allogeneic iPSCs, in which retinal pigment epithelial (RPE) cells derived from allogeneic iPSCs were transplanted into a patient with age-related macular degeneration (AMD) [81,82]. The 2-year follow-up trial demonstrated that the AMD patient’s visual acuity improved and was stable due to the reprogrammed cells, which remained intact. In addition, no serious adverse effects on transplanted iPSC-derived RPE cells were observed [83]. In summary, HLA is a molecule that enables the immune system to distinguish between self and nonself entities, and HLA compatibility is positively correlated with graft survival rates after transplantation [84]. However, the establishment of autologous iPSCs from individual patients validated for clinical use is costly and time-consuming and hampers the standardization of the therapy. Therefore, Japan and the European Union are promoting the use of clinical-grade allogeneic iPSCs, which are established from the peripheral blood mononuclear cells of HLA-matched healthy donors and are less susceptible to immune rejection. The iPSC bank in Japan reported that the stock of their iPSCs was able to cover approximately 32% of the Japanese population in view of HLA matching for clinical use and the targeting of various diseases in 2018 [85].

Global trends in clinical trials including pluripotent stem cells involve ophthalmic diseases, cardiovascular diseases, neurological disorders, metabolic diseases, genetic syndromes, reproductive and urogenital diseases, hematologic disorders, otorhinolaryngologic diseases and defects in the immune system [86]. Additionally, they can be applied for the treatment of spinal cord injury (SCI) [87,88,89], which is considered a refractory traumatic disease. However, with recent advances in stem cell transplantation, the field of regenerative medicine has gained hopeful momentum in developing a novel treatment for this challenging pathology. SCI usually results in permanent disability, and its symptoms include a loss of muscle function, sensation, or autonomic function in the parts of the body served by the spinal cord [87]. Hideyuki O et al. reported the first study to investigate the therapeutic efficacy and safety of human iPSC-neural precursor cell (NPC) transplantation for animal SCI models [90]. In 2018, the Keio University Certified Special Committee for Regenerative Medicine approved the unintegrated human iPSC product for clinically treating SCI patients with the ASIA impairment score A [91].

iPSCs are established from patients across a panoply of diseases, leading to the development of a wide range of cell-based disease models, which are beneficial for understanding the pathogeneses of diseases and facilitating drug discovery. For example, histone deacetylase 4 (HDAC4) is shown to be mislocalized in patient iPSC-derived dopaminergic neurons, which model Parkinson’s disease and cause the downregulation of some critical genes [92]. Alternatively, adoptive immunotherapy with iPSC-derived immune effector cells (dendritic cells (also called DCs), tumor-specific T cells and NK cells) is also applied to cancer treatments [93,94,95,96,97], some examples of which are provided as follows:iPSC-DCs: In the study by Senju S et al., a method was developed to generate DCs from human iPSCs. These iPSC-DCs have the characteristics of original DCs, including the capability of T-cell stimulation, processing and presenting antigens, and producing cytokines [93]. Kitadani J et al. successfully established iPSC-DCs from the fibroblasts of healthy donors, as well as mouse iPSC-DCs from the iPS cell line iPS-MEF-Ng-20D-17 [94], which were derived from C57/BL6 MEFs. They demonstrated the therapeutic potential of mouse iPSC-DCs, in which the carcinoembryonic antigen (CEA) was transduced and expressed in a subcutaneous tumor model using CEA transgenic mice. These findings indicate that genetically modified iPSC-DCs, inducing the expression of CEA, are a promising strategy for the treatment of gastrointestinal cancer.iPSC-T: Adoptive immunotherapy with antigen-specific cytotoxic T lymphocytes (CTLs) represents a potential therapeutic strategy that can reduce tumor development and provide a survival advantage for patients undergoing cancer therapies. Maeda T et al. developed a simple method to generate antigen-specific CD8αβ T cells from the iPSCs of healthy volunteers and demonstrated their therapeutic potential against leukemia [95]. Wilms’ tumor antigen 1 (WT1)-specific CTLs regenerated by this method demonstrated antigen-specific cytotoxic activity in vitro and showed comparable potential to primary CTLs in producing IFNγ and TNFα [94]. When applied in vivo in a xenograft model, these CTLs prolonged the survival of mice bearing WT1-expressing leukemic cells [94]. Recent advances show the potential of the chimeric antigen receptor (CAR)-transduced T-cell immunotherapy for the treatment of a wide variety of diseases. Themeli M et al. demonstrated in clinical trials that CD19 CAR-modified T cells efficiently induce a complete remission in patients with acute or chronic lymphoblastic leukemias and eradicate B-cell malignancies in mice [96].iPSC-NK: The multiple dosing of allogeneic iPSC-NK cell therapy succeeded in treating solid tumors, such as ovarian cancer [97,98]. Hermanson D et al. established iPSCs derived from umbilical cord blood CD34+ cells, UCBiPS7, and derived iPSC-NK cells via spin embryoid bodies [98]. iPSC-NK cells were applied to treat NOD/SCID/γc^−/−^ (NSG) mice, which were inoculated with ovarian cancer cells (MA148), and the median survival improved from 73 to 98 days. Moreover, such iPSC-NK cells were found in the peritoneal cavity of mice and were able to markedly inhibit tumor growth [97], indicating the therapeutic potential of iPSC-NK cells for treating solid tumors.

Although iPSC-Ts provide an alternative cell source for allogeneic T-cell immunotherapy, they appear to have poor outcomes and severe side effects [99,100]. Experimental mice receiving iPSC-T treatment died of tumor relapse and/or graft-versus-host disease (GvHD) [99]. In fact, it is reported that patients may develop cytokine release syndrome (CRS) and/or immune effector cell-associated neurotoxicity syndrome (ICANS) [100]. In addition, although more effective than traditional chemotherapy, T cell therapies are usually costly (approximately USD 373–475,000 per dose), require longer preparation times, and partly depend on the quality of the cell source after leukapheresis [101]. However, NK cells function as allogeneic effectors and do not need to be collected from a patient or a specific HLA-matched donor to reduce GvHD [102]; therefore, it is not necessary to spend much effort preparing autologous NK cells, and off-the-shelf sources of allogeneic iPSC-NK cells may become available. Several trials have demonstrated that 30–50% of patients with refractory or relapsed acute myelogenous leukemia (AML) can achieve complete remission after receiving allogenic NK cells [16] stimulated with cytokines (typically IL-2 or IL-15) [103].

### 2.3. Generation of iPSC-Derived NK Cells

Recently, several generation methods have been described to mass-produce iPSC-NK cells to provide unlimited NK cells for research or clinical applications, and the representative NK cells are summarized in Table 1:PB-iPSCs with OP9: On day 0, iPSCs derived from peripheral blood cells (PB-iPSCs) were cocultured with OP9 cells (a bone marrow stromal cell line) in αMEM with 20% fetal bovine serum (FBS). On day 12, the modified OP9 cell line expressing Notch ligand Delta-like-1 (OP9-DLL1) replaced OP9 cells and was cocultured with the above iPSCs (mainly CD34+) in the presence of the stem cell factor (SCF) and Flt3L, together with IL-7 and IL-15. On day 26, a small population of CD45+CD56+ cells appeared; the CD45+CD56+ cells became the dominant population, with a purity of 99% on day 40. A yield of 7.93 × 10^6^ CD45+CD56+ cells was obtained on day 40 and increased to 15 × 10^6^ cells on day 47 [104].FB-iPSCs with OP9: On day 0, the iPSC cell line from primate skin fibroblasts (FB-iPSCs) was cocultured with OP9 in αMEM containing 20% FBS and supplemented with a basic fibroblast growth factor (bFGF), activin A, vascular endothelial growth factor (VEGF) and CHIR99021. On days 6 and 8, SCF, thrombopoietin (TPO), IL-3 and IL-6 were added to the above culture medium for the differentiation of mesodermal cells to hematopoietic stem cells. On day 10, the floating cells were harvested and cocultured with the modified OP9 cell line expressing Notch ligand Delta-like-4 (OP9-DLL4) in αMEM containing 20% FBS, in addition to IL-7, FLT3L and IL-2 for up to 4 weeks. On day 38, a yield of 1.0–3.5 × 10^6^ iPSC NK cells expressing perforin and IFNγ was obtained [105].CB-iPSC: On day 0, UCBiPS7 iPSCs derived from umbilical cord blood CD34+ cells were seeded in round-bottomed plates for the development of embryoid bodies in a BPEL culture medium (bovine serum albumin, polyvinyl alcohol, essential lipids) containing SCF, VEGF and bone morphogenic protein 4 (BMP-4). During days 8–12, the formed embryoid bodies (EBs) containing CD34+CD43+ cells were directly transferred into flat-bottomed plates, and BPELs were cultured in the presence of IL-3, IL-7, IL-15, SCF and FLT3L. After 28~32 days, iPSC-NK cells were obtained and expanded in RPMI-1640 containing 10% FBS, 1% penicillin/streptomycin and 50 units/mL IL-2 and stimulated with irradiated (10,000 cGy) artificial antigen-presenting cells (aAPCs) (2:1 v/v) upon initiation of culture. The culture medium was changed twice weekly, and iPSC-NK cells could be restimulated with aAPCs every 7 days. The purity of the expanded NK cells almost reached 97% [97].CB-iPSC: On day 0, the iPSC cell lines 409B7 (B7) and CB-A11 (A11), derived from cord blood mononuclear cells, were seeded in iMatrix 511-coated plates (Osaka, Japan) for the development of EBs in an Essential 8 culture medium supplemented with CHIR99021, BMP-4, and VEGF. On day 2, the formed EBs containing CD34+ cells were cultured in an Essential 6 culture medium containing SB431542, SCF, and VEGF. During days 4–12, the formed EBs appeared to contain CD34+, CD43+ and CD45+ hemoangiogenic progenitor cells (HPCs) and were cultured in a Stem Line II medium together with SCF and Flt3L. From day 12 onwards, the cells were cultured in DMEM containing 20% human AB serum or a Stem Line II, in addition to SCF, Flt3L, IL-7 and IL-15. On day 48, the purity of iPSC-NK cells reached 63.10 ± 7.01%~78.23 ± 5.66% [106].

Recently, biotechnology has advanced to the point where NK cells can be generated directly from CD34+ HSCs and iPSCs. However, the different culture methods might give rise to different subsets of NK cells. Therefore, there is an urgent need to standardize the phenotyping protocol because the specific phenotypes may be associated with the function of NK cells, which is critical for the therapeutic applications of NK cells [107].

## 3. Natural Killer Cells (NK cells)

NK cells are a type of lymphocyte (a white blood cell) that was first described in the mid 1970s as an innate immune cell and recently reclassified as a member of the group 1 innate lymphoid cells (ILCs) [108]. Five major groups of ILCs have been defined on the basis of their cytokine production patterns and developmental transcription factor requirements: natural killer (NK) cells, group 1 ILCs (ILC1s), ILC2s, ILC3s and lymphoid tissue-inducer (LTi) cells. ILC1s, ILC2s and ILC3s resemble the corresponding T helper cell subsets (T helper 1 (TH1), TH2 and TH17 cells, respectively) and produce cytokines that shape both innate and adaptive immune responses [109]. They are able to recognize stressed cells in the absence of antibodies and the major histocompatibility complex (MHC), allowing for a much faster immune reaction. NK cells are typically activated by missing MHC class I marker-harmful cells, which play a very important role. Missing MHC class I cells cannot be detected and destroyed by other immune cells, such as T lymphocytes [109,110,111,112]. 

### 3.1. The Physiological Conditions in NK Cells

NK cells differentiate from and mature in the bone marrow (BM), lymph nodes (LNs), spleen, tonsils, and thymus and then enter the circulation. In humans, they are typically characterized as CD56+CD3^−^ lymphocytes and can be broadly categorized into two subpopulations based on the level of CD56 and CD16 (Fc receptor FcR III) expression: CD56^bright^/CD16^neg^ cells and CD56^dim^/CD16^pos^ cells (Figure 3). Previous studies have suggested that CD56^bright^/CD16^neg^ cells are immature precursors of mature CD56^dim^/CD16^pos^ cells [113,114,115]. The majority of PB-NK cells are CD56^dim^/CD16^pos^ cells, which are highly cytotoxic against target cells. In contrast, approximately 2–10% of PB-NK cells are CD56^bright^/CD16^neg^ cells, which have a low cytotoxic activity while displaying a high capacity to produce immune regulatory cytokines (such as IFNγ, TNF, and GM-CSF) [116], interacting with dendritic cells and T-cell polarization to participate directly in adaptive immune responses [117]. The unique characteristic of the metabolism of NK cells is that they do not use glutamine as a fuel to drive oxidative phosphorylation (OXPHOS). In fact, ATP production is primarily fueled by glucose. The inhibition of OXPHOS or glycolysis in NK cells significantly impairs IFNγ production in a short time [111].

Mature NK cells are known as CD56^dim^/CD16^pos^ cells that can trigger antibody-dependent cellular cytotoxicity (ADCC), which is known as one of the important functions of NKs. In fact, several different mechanisms are involved in the NK-mediated lysis of target cells [118,119], including (1) the release of cytoplasmic granules containing perforin and granzymes, (2) the production of IFN-γ, (3) the expression of FasL and TRAIL, and (4) the expression of ADCC. For example, in tumor cell therapy, the type III Fc-gamma receptor (FcγR), also known as CD16, on NK cells recognizes the Fc portion of antibodies bound to tumor cells and triggers the cell death of tumor cells through ADCC. This antibody may be a monoclonal antibody (mAb), preferentially of class IgG1 or IgG3 since these two antibodies are able to link different FcγRs [120]. Over 100 monoclonal antibodies (mAbs) on the market involving several mechanisms of action are used as tumor cell therapeutics [121], in addition to NK cell-mediated ADCC, including checkpoint inhibitors, targeting radiation, blocking cell growth, inhibiting neovascularization and inducing leukocyte effector functions [120,122,123]. 

### 3.2. NK Cell Education

NK cell activity is tightly regulated by a complex interplay between activating and inhibitory receptors that prevent the killing of normal autologous cells expressing an appropriate level of all self-HLA alleles and low/negative levels of ligands for non-HLA-specific activating receptors (aNKRs) [124]. The most frequently described activating receptor is natural killer group 2D (NKG2D, a transmembrane protein), which belongs to the natural cytotoxic receptor (NCR) family. The other family members include NKp46, NKp30, and NKp44, and the leukocyte adhesion molecule DNAX accessory molecule-1 (DNAM1, also called CD226). These receptors, expressed largely on NK cells, are potent inducers of NK cell cytotoxicity, and are crucial for NK cell-mediated tumor apoptosis. It is known that the expression of NKp44 is induced upon NK cell activation, while NKp46 and NKp30 are expressed on both resting and activated NK cells [125]. 

There are two main inhibitory receptors, the killer immunoglobulin-like receptor (KIR) family, which can bind HLA-class I, and the heterodimeric receptors CD94-NKG2A/B, which recognize HLA-E [124,126,127]. In malignancies, activating killer cell immunoglobulin-like receptors (KARs) are often decreased, while the expression of the most prominent inhibitory NK cell receptors, KIRs and CD94/NKG2A, may occasionally increase [128,129]. Remarkably, NK cell activation is determined by the balance of inhibitory and activating receptor stimulation [129,130]. Indeed, in the absence of inhibitory interactions, NK cells kill target cells and produce cytokines in great quantities [128]. By secreting large amounts of cytokines and chemokines, mature NK cells can not only directly kill target cells but can also elicit other immune cells, including monocytes, DCs, and T cells. Similarly, naïve NK cells can also be activated by different proinflammatory cytokines, such as IL-2, IL-15, IL-18 or IL-21, which stimulate NK cell survival and proliferation, as well as upregulating the expression of activating receptors and enhance NK cell cytotoxicity [109,110].

NK cells can recognize tumor cells and have antitumor and antimetastatic potential in cancerous cells [131]. The chemoattractant/receptor axes appear to control tumor-infiltrating NK cell migration, activation, survival, and persistence in the tumor microenvironment [132]. In the context of NK cells, chemokine receptors of note include CCR2, CCR5, CCR7, CXCR3, and CX3CR1 [133]. CCR2 and CCR5 regulate the migration of tumor-associated monocytes and macrophages. The cytolytic activity of NK cells is simultaneously augmented by CCL2 [134]. High levels of CCL2 correlate with increased monocyte and macrophage recruitment and appear to be an indicator of an adverse prognosis in patients with breast, ovarian, gastric, and esophageal carcinomas [135,136]. The presence of tumor-infiltrating NK cells confers a favorable outcome in many tumors. However, due to nutrient and oxygen deprivation, a higher concentration of tumor-derived metabolism causes the metabolic impairment of NK cells, which ultimately limits their effector functions [137].

### 3.3. NK Cell-Based Therapy in Tumors

Many trials of adoptive NK cell-based immunotherapy have been performed over the past decade and growing clinical and experimental evidence highlights the clear and direct role of NK cells in controlling human cancer development and/or progression. Epidemiologic studies indicate that a reduced function of NK cells is related to cancer incidence [107]. Endogenous NK cells in cancer patients usually have impaired function because of the alteration of the receptor repertoire in the cells. Therefore, the primary method in immunotherapy treatments is to “push” for immune activation by including additives such as cytokines (IL-2 therapy [138], IL-15 therapy [139] and TGFβ inhibitors [140]) and antibodies that help to modulate the mechanisms that improve the quantity and/or quality of the antitumor immune response. For example, some immunomodulatory drugs targeting NK cells, such as lenalidomide and pomalidomide, are approved for the treatment of multiple myeloma, mantle cell lymphoma and a subset of myeloid-derived suppressor cells (MDSCs). These drugs induce cell cycle arrest and apoptosis in tumor cells. Moreover, they increase NCR expression, expand NK cell populations and increase the immune cell recognition of tumor cells in various models. Lenalidomide, for instance, is shown to decrease the immunosuppressive activity of MDSCs and regulatory T (Treg) cells and to increase NK cell cytotoxicity and IFNγ production. NK cell dysfunction in patients with chronic myeloid leukemia (CML) is associated with immune evasion and disease progression, but tyrosine kinase inhibitor (TKI) treatment can restore NK cell numbers and functions [141].

However, NK cell-mediated control of large solid tumors is usually inefficient, although tumors often express large amounts of activating ligands and low levels of inhibitory ligands, presumably due to tumor escape through the alteration of NK cell function and resistance. Currently, the augmentation of the receptor affinity and activation of ADCC and accessibility to the tumor site, in combination with other molecules and immune-modulatory strategies, such as radiotherapy and checkpoint blockade, might be good targets for NK cell-mediated attack. A new type II glycoengineered anti-CD20 mAb with increased FcγR III binding and ADCC is represented by obinutuzumab (GA101). GA101 induces NK cell activation regardless of the inhibitory immunoglobulin-like receptor (KIR) expression. Furthermore, its activity is not adversely modulated by KIR/HLA [120]. Monalizumab (IPH2201), an anti-NKG2A checkpoint inhibitor, has been evaluated in clinical trials in ovarian cancers, head and neck cancers, advanced malignancies and chronic lymphocytic leukemia (CLL). Its use might represent a novel approach in NK-based immunotherapy, not only by enhancing the cytotoxic potential of the cells but also by potentially proposing a role for them in ADCC augmentation [120].

Conversely, many studies have described the engineering of NK cells with CARs to improve the killing of solid tumors, and clinical trials utilizing CAR-expressing NK cells for the treatment of both hematological malignancies and refractory solid tumors have been initiated (ClinicalTrials.gov: NCT00995137, NCT01974479, NCT02839954, NCT02892695, NCT02742727, and NCT02944162, https://clinicaltrials.gov/, accessed on 29 July 2021). Most of these studies use CARs designed for T cells that are expressed in NK cells. However, due to the limitations of CAR-T cell therapy and the patient’s specific clinical and disease-related features, a decision was made to take an alternative approach to using CAR natural killer (NK) cells that might circumvent these problems [141,142]. Even though allogeneic iPSC-NK cell therapy requires multiple doses for the treatment of solid tumors, genetic modifications, CAR-NK can further improve the specificity, strength, and efficacy of iPSC-CAR NK cell therapies [141].

Indeed, there is an increasing interest in iPSC-NK therapies due to their ability to address the supply chain bottlenecks associated with primary and cell line NK therapies (Table 2). The advantages of iPSCs include their ease of generation from accessible sources such as fibroblasts or peripheral blood, the retention of pluripotency during expansion, and the capacity for long-term storage. The FDA has already approved a phase I clinical trial to investigate Fate Therapeutics’ off-the-shelf iPS-NK product, FT500, representing the first FDA-approved clinical investigation of an iPS-derived cell product in the USA. To generate this product, NK cells were developed from a clonal master iPSC line cell bank. FT500 in combination with checkpoint blockade therapy was utilized to treat adults with advanced solid tumors (ClinicalTrials.gov: NCT03841110). FT516, engineered to include a high affinity, noncleavable CD16 (hnCD16) Fc receptor, was designed against hematological malignancies (ClinicalTrials.gov: NCT04023071). In the preclinical FT538 study, NK cells were engineered by the knockout of the CD38 receptor and knock-in of the high-affinity, noncleavable CD16 receptor and the fused IL-15 receptor. The cells were used in combination with daratumumab (anti-CD38) monoclonal antibody to treat multiple myeloma [101]. 

As of the data cutoff date of April 16, 2021, the encouraging outcome for pateints in both trials of FT516 and FT538 was observed. Four patients in FT516 (*n* = 9) and one in FT538 (*n* = 3), respectively, achieved an objective response with complete leukemic blast clearance in the bone marrow. None of them observed dose-limiting toxicities and no events were of any grade of CRS, ICANS, or GvHD. Recently, a third-generation anti–GPC3-CAR 28bbz has demonstrated safety and efficacy in disseminating ovarian tumors [143]. Being engineered with chimeric antigen receptors, the iPSC-NK cells appears to be more effective at binding to cancer-specific antigens, enhancing the therapeutic efficacy for solid tumors, and providing perspectives for its clinical uses.

## 4. Conclusions

Despite their potent antitumor activity, NK cells face substantial challenges that hinder their efficacy, such as difficulty in obtaining a large quantity of NK cells, expansion to the clinical scale ex vivo, and the ability to sustain in vivo survival and activity. These challenges can be resolved with the new iPSC-NK tools and genetic engineering approaches. However, the iPSC safety profile, particularly in relation to tumorigenic potential, remains to be understood in sufficient detail for clinical translation. Undoubtedly, NK cells are powerful tools in the armamentarium against cancer due to their direct cytolytic activity against tumor cells. Recently, more treatments using NK cells have shown good results in hematological malignancies and promise a new paradigm for the treatment of solid tumors. Although NK cells have been overlooked in the field, we can target tumor-induced NK cell inhibition to promote the maximum antitumor effect in NK cell-based immunotherapies.

## Figures and Tables

**Figure 1 biomedicines-09-01323-f001:**
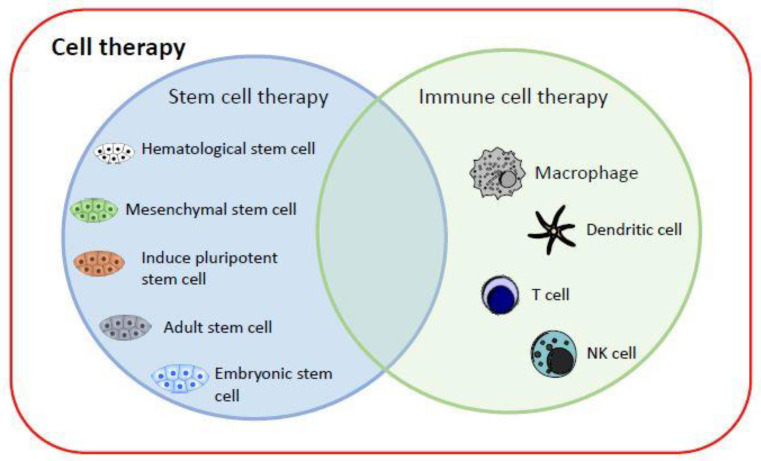
Current cell therapy categories. The major cell therapy categories include stem cell therapy and immune cell therapy. They can replace and repair damaged cells, tissues, and organs in humans, or kill target cells.

**Figure 2 biomedicines-09-01323-f002:**
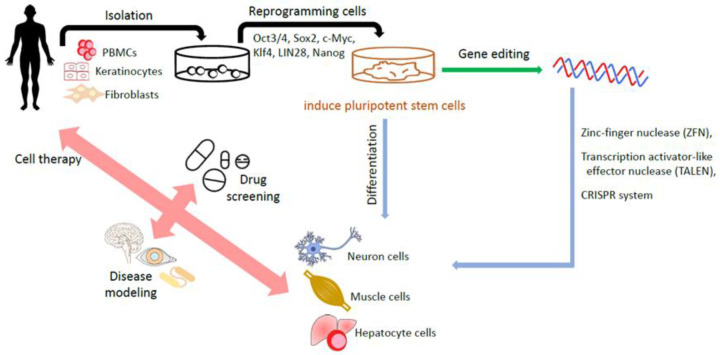
Generation of human iPSCs and their applications. The somatic cells including fibroblasts and isolated from a patient are reprogrammed into iPSCs by transduction with the reprogramming factors. The gene editing technology helps to create iPSCs-based disease cell model. Eventually, iPSCs, with or without edited modifications, are differentiated into various target cells for disease modeling, drug screening, and cell therapy.

**Figure 3 biomedicines-09-01323-f003:**
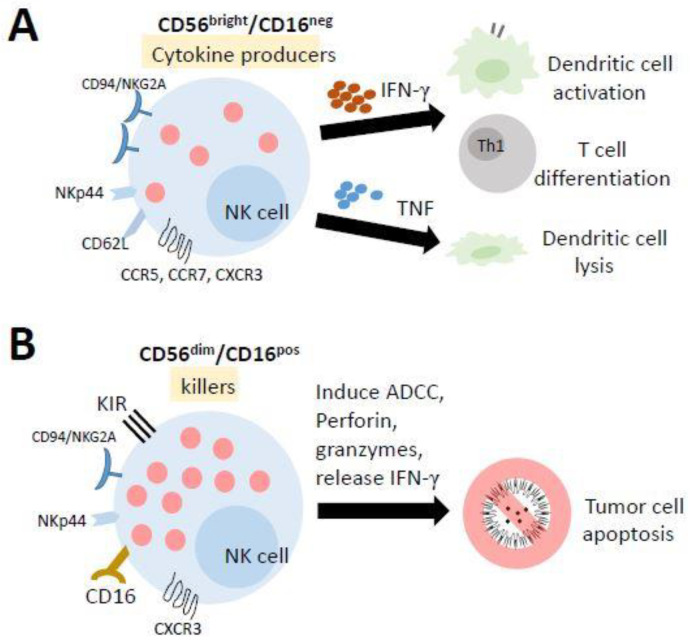
The mature human natural killer (NK) cells and their functions. NK cells differentiate from and mature in the bone marrow, lymph nodes, spleen, tonsils, and thymus and then enter the circulation. (**A**) Most of the cells in the lymph nodes (LN) and tonsils have a lower cytotoxic activity while displaying a high capacity to produce several cytokines. They promote the activation of dendritic cells and the polarization of Th1 cells by releasing IFN-γ, while TNF production results in the lysis of immature DCs. (**B**) The cells in the peripheral blood (PB) and spleen are highly cytotoxic against target cells. They can trigger antibody-dependent cellular cytotoxicity (ADCC) to kill target cells.

**Table 1 biomedicines-09-01323-t001:** Generation methods for iPSC-derived NK cells.

	Primary Differentiation			Lymphoid Commitment			Yield	Reference
	Medium	Cytokine	Culture Day	Medium	Cytokines	Culture Day	(per 1 × 10^6^ iPS cells)	
1	αMEM + 20% FBS	-	12	αMEM + 20% FBS	SCF, Flt3L, IL-7, IL-15	47	15.0 × 10^6^	[104]
2	αMEM + 20% FBS	bFGF, activin A, VEGF, CHIR99021	10	αMEM + 20% FBS	Flt3L, IL-7, IL-2	38	1.0~3.5 × 10^6^	[105]
3	BPEL (APEL + 10% FBS)	SCF, VEGF, BMP-4	11	BPEL	SCF, Flt3L, IL-3, IL-7, IL-15	28~32	>97%	[98]
4	Essential 8	BMP-4, CHIR99021, VEGF	0–2	DMEM + 20% human AB-serum or Stem line II	SCF, Flt3L, IL-7, IL-15	48	63.10 ± 7.01%~78.23 ± 5.66%	[106]
	Essential 6	SCF, SB431542, VEGF	2–4					
	Stem line II	SCF, Flt3L	4–12					

ND: non data. CHIR99021 is an aminopyrimidine derivative that is an extremely potent glycogen synthase kinase (GSK) 3 inhibitor, inhibiting both GSK3β and GSK3α. SB431542 is a selective and potent inhibitor of the TGF-β/Activin/NODAL pathway that inhibits ALK5, ALK4 and ALK7 by competing for the ATP binding site. Both are commonly used to maintain human and mouse stem cell lines.

**Table 2 biomedicines-09-01323-t002:** iPSC-NK cell therapy for cancers.

Pre-Clinical Research				
Disease target	Strategies	Outcome		References
Ovarian cancer	Multiple dose, IL-2 stimulated	The median survival improved from 73 to 98 days		[97]
Cell line (K562, SKOV3, SW480, HCT-8, MCF7, SCC-25)	IL-2 stimulated	Efficiently killed all tested cancer cell lines (*p* < 0.5)		[105]
Ovarian cancer	Targeting Mesothelin, engineered with chimeric, NKG2D-CAR-iPSC-NK	NKG2D-CAR-iPSC-NK cells displayed in vivo function similar to NKG2D-CAR-iPSC-T cells		[141]
Hematological cancers, Hepatocellular carcinomas, Ovarian cancer	Tetravalent bispecific trifunctional antibody targeting GPC3, NKp46-CAR-iPSC-NK-EGFR	Effectively suppressed GPC3-expressing tumor growth in vitro and in vivo and confirmed the therapeutic quality and safety of the final product		[142]
**Clinical trial**				
NCT number	Disease target	Phase	Start date	Affiliation
NCT03841110	Advanced solid tumors Lymphoma, Gastric cancer, Colorectal cancer, Head and neck cancer, Squamous cell Carcinoma EGFR positive solid tumor, HER2-positive breast cancer, Hepatocellular, Small cell lung cancer, Renal cell carcinoma, Pancreas cancer, Melanoma, NSCLC, Urothelial carcinoma, Cervical cancer, Microsatellite instability, Merkel cell carcinoma	I	15-Feb-19	Fate Therapeutics
NCT04023071	Acute myelogenous leukemia, B-cell lymphoma	I	4-Oct-19	Fate Therapeutics
NCT04614636	Multiple myeloma Relapsed/Refractory acute Myeloid leukemia, Acute myelogenous leukemia	I	4-Nov-20	Fate Therapeutics

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
