# Peer review of "An Alternative Cell Therapy for Cancers: Induced Pluripotent Stem Cell (iPSC)-Derived Natural Killer Cells"

_biomedicines, 2021, doi:10.3390/biomedicines9101323_

Round 1
Reviewer 1 Report
Thank you for the opportunity to review this manuscript. This manuscript is a comprehensive review of the involvement of inflammasome in the pathogenesis of COVID-19. This manuscript is a comprehensive review of cancer immunotherapy using iPSC-derived NK cells. This manuscript is well written, and the information here might be helpful for readers. However, few concerns need to be improved before publication. Please see my specific comment stated below.
Major Comments
- The major weakness of this review is the lack of information about iPSC-derived NK cell-based cancer therapy using both animal models and clinical trials for human usage. Therefore, please provide more descriptions of this information.
- Another weakness of this manuscript is the lack of description of basic sciences of NK cells such as NK cell receptors, the regulation of NK cell activity, and NK cell-mediated killing. Therefore, please provide more descriptions of this information.
Reviewer 2 Report
In this review paper, Hsu et al. try to give an overview of iPSC and natural killer (NK) cells, with a bridge between both and details about the application of iPSC-derived NK cells in cancer immunotherapy (cellular therapy). The problem is that the NK cell part is not very original, as tons of similar papers are existing in the literature. In contrast, the iPSC part comes up with unnecessary details that have no place in a review, but better in a Methods paper. To sum up, this manuscript adds little to nothing to the literature in the field. It should be entirely rewritten by shortening the iPSC part, taking out the majority of the NK cell part and focusing on lines 471-513 that should be developped, as these are the true topic of the article according to the title.
In addition, there are a few more points to be corrected/clarified:
1) English language could be edited to a better level. Style is not exceptional.
2) In figure 1, macrophages could be added as cell therapy effectors.
3) Lines 90-91: what is the sense of this sentence? NK92 is a tumor cell line and not iPSC-derived.
4) Line 93: chemical signals?
5) Line 121: ectoderm appears twice.
6) Lines 138 and 140: again twice the same affirmation.
7) Figure 2: PBMC are not mentioned in table 1.
8) Lines 155 and 158: twice the same sentence!
9) Line 295: CHIR99021 (and SB131542) should be briefly explained.
10) Line 310: what is this artificial antigen presenting cell line?
11) Chapter 2.3 should be shortened dramatically as not interesting in a review article; table 2 is enough.
12) Line 374: ADCC is one, but not the most important function of NK cells.
13) Lines 377-380: this statement is wrong. All NK cell activation does not go through CD16. Please clarify.
14) "The antiviral role of of ADCC in clearing tumors". What do the authors want to claim here?
15) Lines 373-389: overall confusing and should be rewritten.
16) Lines 412-413: this does not make any sense.
17) In figure 3, replace TRAIL by perforin and granzymes because the latter are much more important physiologically.
18) Line 450: NK cell based and not base.
